



# Predictive and Stochastic Reduced Order Modelling of Wind Turbine Wake Dynamics

Søren Juhl Andersen[1,*] and Juan Pablo Murcia Leon[2,*]

[1]Department of Wind and Energy Systems, Technical University of Denmark, Anker Engelunds Vej 1, 2800 Kgs Lyngby, Denmark
[2]Department of Wind and Energy Systems, Technical University of Denmark, Frederiksborgvej 399, 4000 Roskilde, Denmark

**Correspondence:** Søren Juhl Andersen (sjan@dtu.dk)

**Abstract.**

This article presents a reduced order model of the highly turbulent wind turbine wake dynamics. The model is derived using a database of Large Eddy Simulations (LES), which cover a range of different wind speeds. The model consists of several sub-models: (1) dimensionality reduction using Proper Orthogonal Decomposition (POD) on the global database, (2) projection in

modal coordinates to get time series of the dynamics, (3) interpolation over the parameter space enables prediction of unseen cases, and (4) stochastic time series generation to generalize the modal dynamics based on spectral analysis. The model is validated against an unseen LES case in terms of the modal time series properties as well as turbine performance and aero-elastic responses. The reduced order model provides LES accuracy and comparable distributions of all channels. Furthermore, the model provides substantial insights of the underlying flow physics, and how these change with respect to the thrust coefficient

$C_T$ and whether the model is constructed for single wake or deep array conditions. The predictive and stochastic capabilities of the reduced order model can effectively be viewed as a generalization of LES for statistically stationary flows, and the model framework can be applied to other flow cases than wake dynamics behind wind turbines.

## 1 Introduction

Wind turbines in large wind farms are subject to highly turbulent inflow as they operate in the wake of upstream turbines. The inherently complex and dynamic inflow determines the performance and operation of the individual turbines in terms of both reduced power production and increased loads. Understanding and accurate modelling of wind turbine wakes is therefore paramount for improving wind farm design and operation (Veers et al., 2022). Numerous models have been developed in the last decades aimed at simplifying the physics in order to address different aspects of wind farm flow, but it remains a challenge

to develop fast and accurate dynamic flow models that correctly captures the turbulent wake flows on its wide range of scales in both time and space (Meneveau, 2019; Veers et al., 2019; Porté-Agel et al., 2020).



The dynamics of the turbulent wake flows are particular challenging to model, and hence the uncertainty of estimating *e.g.* damage equivalent loads are significant. Various stochastic models (Veers (1988); Mann (1994, 1998); Sørensen et al. (2002)) can be used to generate time series of wind velocity fluctuations to match the turbulent flow statistics, although it will not directly include the influence of the wakes. Stochastic turbulence is also used to drive the Dynamic Wake Meandering (DWM) model Larsen et al. (2007, 2008), which combines contributions from the time-averaged wake deficit, large scale meandering assumed to originate from the largest atmospheric scales, and small scale added turbulence. However, DWM assumes a separation of scales by linearly combining the effects of its three components.

Large eddy simulations (LES) solves the non-linear Navier-Stokes equations and can therefore elucidate the turbulent wake flow and the performance of large wind farms, see *e.g.* (Wu and Porté-Agel, 2013; Stevens and Meneveau, 2017; Allaerts and Meyers, 2018; Andersen et al., 2020), but the computational costs are very high. Additional insights can be gained by applying different data-driven methods on the LES generated data, for instance Proper Orthogonal Decomposition (POD), which provides an optimal linear subspace in terms of turbulent kinetic energy. POD (and similar dimensional reduction techniques) has been used extensively to analyse turbulent flows generally, and specifically in the context of wind farm flows it has for example been applied to reveal the underlying mechanisms of energy entrainment and wake recovery (VerHulst and Meneveau, 2014; Newman et al., 2014; Andersen et al., 2017; Cillis et al., 2020). POD can also be used to construct reduced order models (ROM) of turbulent flows by truncating the number of POD modes use to represent a flow. Reduced order models of wind turbine wakes have typically focused on reconstructing single flow cases, *e.g.* Andersen et al. (2014); Debnath et al. (2017). Others have intended to expand ROM's application beyond reconstruction to include stochastic flow generation or forecasting (Bastine et al., 2018; Hamilton et al., 2018; Moon and Manuel, 2021).

However, as summarized by Meneveau (2019) the past attempts have failed in utilizing POD to truly develop predictive and stochastic reduced order models. The motivation of this work is to resolve these past shortcomings by creating a reduced order model that fulfills three requirements:

1. Stochastic, *i.e.* the ability to generate different flow realizations with accurate statistics.

2. Predictive, *i.e.* the ability to predict wake dynamics for input parameters different than those used to develop the POD modes.

3. Speed. A computational fast dynamic wake model can be used to generate numerous flow realizations and therefore can predict statistical distributions of the quantities of interest.

The proposed modelling framework consist of a reduced order model based on POD modes derived from LES data covering a parameter space. Stochastic flow generation is achieved using multivariate spectral analysis. Finally, the wake model is generalized over the parameter space in order to predict unseen cases. However, the full parameter space influencing wind farms flows is large. Atmospheric boundary layer flows can be parameterized in terms of *e.g.* roughness, geostrophic wind, and various temperature effects, which gives various combinations of wind speeds, shear, veer, and turbulence intensities over the turbine. Similarly, the wind farms are described by rotor size, hub height, turbine spacing and operation etc. In this article





we present a reduced order model covering a significantly reduced parameter space, which covers only the range of the most important parameter and maintains fixed values for all other parameters, *e.g.* atmospheric turbulence intensity, spacing, rotor size. The wake dynamics are primarily governed by the relative turbine forcing on the flow, *i.e.* the thrust coefficient $C_T$ (van der Laan et al., 2020). Therefore, $C_T$ essentially indirectly encapsulate the effects of e.g. atmospheric turbulence intensity and turbine spacing, as increases will lead to increased wake recovery and therefore typically increases in $C_T$. Hence, the reduced order model is developed to cover the entire operational $C_T$-range as the only governing parameter.

The model framework is applied to the inflow for two different turbines in a long row in order to demonstrate the universality of model by covering two distinct cases: single wake flow and deep array wind farm flow. The model is validated in terms of the probability distributions of power and loads of the turbines operating under waked conditions. It is important to note, that the elementary questions and challenges addresses here are not unique for wind farm flows, but essentially relates to fast and accurate modelling of turbulent flows in general, which remains a fundamental research topic [1].

## 2 Methods

### 2.1 Predictive and Stochastic Reduced Order Model (PS-ROM)

The proposed modelling framework consists of 5 steps as depicted in Figure 1:

1. A database of LES flow simulations performed for different flow-cases characterized by having different values in the parameter space. Possible transients and non-stationary behavior are removed on each simulation in the database in order to ensure stationarity and ergodicity.

2. Global Proper Orthogonal Decomposition (POD) is used to obtain global modes of the flow fluctuations. This step has the main objective of reducing the number of dimensions needed to represent the flow across the parameter space.

3. Projection of the flows in the database into modal coefficient time-series is used to describe the dynamics of the fluctuating flow in POD modes. The complex cross-spectral density matrix (CSD) across all the modal time-series is used to describe the second order statistics of the flow in terms of cross-covariance functions across different modal time series.

4. The prediction aspect of PS-ROM relies on the interpolation of the CSD within the parameter space.

5. Stochastic realizations of the dynamic wake flows are obtained using classical complex CSD time-series generation methods, which relies on building a transfer function to "color" a random realization of white noise into time-series with the desired cross-spectra.

---

[1]https://www.claymath.org/millennium-problems/navier%E2%80%93stokes-equation





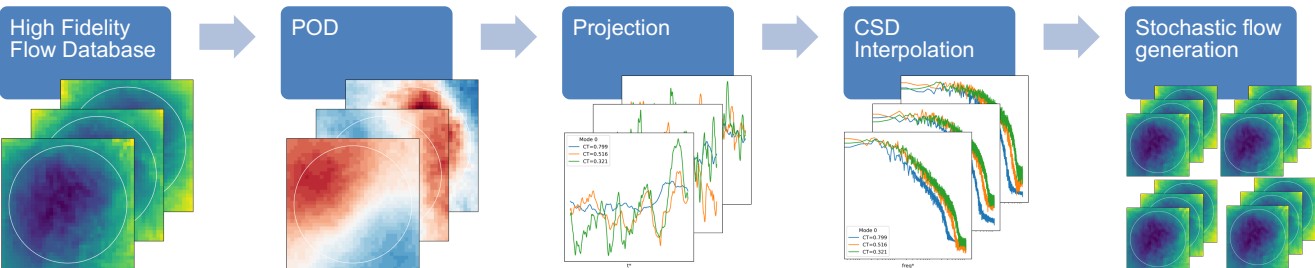

**Figure 1.** Five steps of the Predictive Stochastic Reduced Order Model.

## 2.2 Flow Solver and Turbine Modelling

The database of wind farm flow cases is constructed using the incompressible 3D flow solver EllipSys3D (Michelsen (1992, 1994); Sørensen (1995)). EllipSys3D solves the discretized Navier–Stokes equations in general curvilinear coordinates using a block structured finite volume approach in a collocated grid arrangement. The pressure correction equation is solved with an improved version of the SIMPLEC algorithm (Shen et al., 2003) and pressure decoupling is avoided using the Rhie-Chow interpolation technique. The convective terms are discretized using a third order QUICK scheme, and a second order accurate implicit method is used for time stepping using sub iterations. LES is employed, which applies a spatial filter on the Navier–Stokes equations, where the smaller scales are modeled through a sub-grid scale (SGS) model, which provides the turbulence closure. Here, the SGS model by Deardorff is used (Deardorff, 1980).

The turbines are modelled using the actuator disc method, which imposes body forces in the flow equations (Mikkelsen, 2003). Initially, the velocities are passed from EllipSys3D to Flex5 (Øye, 1996), which computes the forces and deflections through a full aero-servo-elastic computation, and transfers these back to EllipSys3D (Hodgson et al., 2021).

## 2.3 Proper Orthogonal Decomposition (POD)

Proper Orthogonal Decomposition (POD) is a classic technique for dynamic flow analysis, which decomposes a turbulent flow into modes of spatial variability and where the orthogonal modes are optimal in terms of capturing the variance of the fluctuating flow, see Lumley (1967); Berkooz et al. (1993). This article introduces a modified approach for decomposing multiple flow-cases over a parameter space into global POD modes.

Two pre-processing steps are required before storing new flow simulations into the flow database with the purpose of applying global POD. Firstly, the flow is normalized by a reference wind speed, $V_0$. Secondly, the flow field in each individual simulation is decomposed into temporal mean and fluctuations.

$$\mathbf{V}(t) = \overline{\mathbf{V}} + \mathbf{V}'(t) = V_0 \left( \overline{\mathbf{V}^*} + \mathbf{V}'^*(t) \right) \tag{1}$$



The global POD consists of applying the snapshot POD formulation (Sirovich, 1987) to normalized velocity fluctuations over multiple flow simulations. These two pre-processing steps have the objective of minimizing the biases towards a single flow-case by ensuring similar range of normalized fluctuating wind speeds.

The normalized velocity fluctuations $\mathbf{V}'^*(t)$ of all the three components: $u, v, w$ and for all points in space are aggregated as column vectors for all $N_t$ time steps from the $N_c$ flow-cases into the snapshot matrix, $\mathbf{M}$. All three velocity components are required in the POD to correctly capture the flow physics (Iqbal and Thomas, 2007). In general, $N_t$ does not have to be identical for all flow-cases, nor does it have to include all the time-steps simulated, but for simplicity all cases are given equal weight (same number of snapshots). Therefore, the total number of instantaneous snapshots is $N_I = N_t \times N_c$ and the snapshot

matrix is built by concatenating the single column vector snapshots: $\mathbf{M} = [\mathbf{V}'^*_{1,1} \ldots \mathbf{V}'^*_{1,N_t} \ldots \mathbf{V}'^*_{N_c,1} \ldots \mathbf{V}'^*_{N_c,N_t}]$. The $N_I \times N_I$ auto-covariance matrix is then computed as $\mathbf{R} = \mathbf{M}^T \mathbf{M}$, and the eigenvalue problem $\mathbf{R}\mathbf{G} = \mathbf{G}\mathbf{\Lambda}$ is solved, where $\mathbf{\Lambda}$ is a matrix of real and positive eigenvalues and $\mathbf{G}$ is a matrix of orthonormal eigenvectors $\mathbf{G} = [\mathbf{g}_1 \ldots \mathbf{g}_{N_I-1}]$, herein denoted as the global spatial POD modes. Note, how the dimension is reduced by one, as the mean was originally subtracted to apply POD only on the fluctuations; and that the orthonormality of the global modes is given using standard inner product across all

flow components: $\langle \mathbf{g}_i, \mathbf{g}_j \rangle = \delta_{ij}$.

A reduced order representation of the flow can therefore be constructed as a combination of a limited number ($K$) of POD modes and their corresponding modal time-series ($\phi_i(t)$):

$$\mathbf{V}(t) \approx \overline{\mathbf{V}} + \sum_{i=1}^{K} \mathbf{g}_i \, \phi_i(t) \tag{2}$$

The challenge in building generalized reduced order models out of a database of flow simulations is then to build models that

are stochastic (different realizations) and predictive (across parameter space) for the modal time-series, as the global POD modes are maintained across the parameter space.

### 2.4   Projection into modal time-series and cross-spectral density (CSD)

The modal time-series for a given flow-case are obtained by projecting the fluctuating flow into the global POD modes using a standard inner product:

$$\phi_i(t) = \langle \mathbf{V}'(t), \mathbf{g}_i \rangle \tag{3}$$

The resulting multivariate modal time-series, $\phi(t)$, are a multivariate random-process with zero-mean and characterized by the complex cross-spectral density matrix (CSD). The CSD of the modal time-series for each flow-case simulation, available in the database, is estimated and smoothed using Welch's averaged windowed periodogram method (Welch, 1967). CSD ($\mathbf{S}(f)$) is defined as the Fourier transfer of the cross-covariance function, and is a complex-number Hermitian matrix at a given

frequency. The imaginary part of the CSD is important because it captures the phase-shift or lags across the different frequency components of the different modal time-series.





## 2.5 Stochastic Time Series Generation

A stochastic representation of the fluctuating flow can be built using classic multivariate Gaussian process spectral theory and/or turbulence field generation (Shinozuka and Jan, 1972; Veers, 1988; Mann, 1994; Sørensen et al., 2002). These methods
can be applied for generating random multivariate fields based on the spectral representation when the time-series are stationary and ergodic.

The stochastic multivariate time series generation consists in decomposing the complex cross-spectral density into a lower triangular matrix, see eq. 4, where $^H$ denotes the Hermitian operator (or complex conjugate transposed). Several methods of matrix decomposition can be used to approximate $\mathbf{H}$ such as Choleski decomposition, Jacobi method etc. This article uses LDL
decomposition, because it by-passes the problems that arise in Choleski decomposition when the CSD matrix is numerically not positive defined, *i.e.* when it has null eigenvalues due to rounding errors.

$$\mathbf{S}(f) = \mathbf{H}(f)\mathbf{H}^H(f) \tag{4}$$

A realization of white noise, $\mathbf{N}(f)$, can be generated by sampling independently uniformly distributed phases, $\Phi$, and the $\mathbf{H}(f)$ matrix is applied as an operator that "colors" the white noise to have the desired cross-spectral properties. Finally, the
realization of the modal time-series, $\tilde{\phi}(t)$, will be obtained by applying the inverse fast Fourier transformation (IFFT):

$$\tilde{\phi}(t) = \mathrm{IFFT}(\mathbf{H}(f)\mathbf{N}(f)) = \mathrm{IFFT}(\mathbf{H}(f)[e^{\mathbf{i}\Phi}]) \tag{5}$$

## 3 Prediction across parameter space

A predictive model of the dynamic flow contains two components: First, a mean flow surrogate ($\hat{\bar{\mathbf{V}}}(\theta)$) that predicts the mean flow for a given parameter ($\theta$). This article will assume that the mean flow surrogate is already available, or that the mean
flow can be simulated either by using RANS-CFD or an engineering wake model. Second, a CSD surrogate that predicts the change of the modal spectra across the parameter space, $\hat{\mathbf{S}}(f, \theta)$. The spectral surrogate can be used in the stochastic time-series generation algorithm to produce a realization of the modal time-series for the new parameter: $\hat{\phi}_i(t, \theta)$. The final flow prediction in the PS-ROM is then given by:

$$\mathbf{V}(t,\theta) \approx \hat{\bar{\mathbf{V}}}(\theta) + \sum_{i=1}^{K} \mathbf{g}_i \hat{\phi}_i(t,\theta) \tag{6}$$





## 4 Results

### 4.1 Flow Database

An aligned row of 14 turbines are modelled for different atmospheric conditions to form a database of turbulent wake flows. The inflow, wind farm and cases constituting the database are defined in the following.

**Inflow**

Initially, a precursor simulations is run to mimic the atmospheric boundary layer (ABL), which serves as inflow to the wind farm simulations. The domain size for the precursor simulation is $2880m \times 1440m \times 960m$ discretized by $576 \times 288 \times 320$ grid cells, corresponding to a grid resolution of $5m \times 5m \times 3m$ in the streamwise ($x$), lateral ($y$), and vertical ($z$) direction. The precursor is performed for neutral atmospheric conditions and driven by a constant pressure gradient over flat terrain, and with cyclic boundary conditions in the horizontal directions. The cyclic boundaries in the streamwise direction have been shifted laterally to prevent spanwise locking of large turbulent structures (Munters et al., 2016) and to reduce the influence of the domain. The initial precursor is simulated with a roughness of $z_0 = 0.05m$ and a friction velocity of $u_* = 0.4545m/s$. The ABL flow was developed for $82,600s \approx 22.94 hours$ to converge the statistics, before inflow data is extracted for the following $28,800s$.

The neutral ABL precursor corresponds to a rough-wall boundary layer for high Reynolds numbers, and therefore the precursor flow can be re-scaled to create a number of different inflow conditions, see Castro (2007). The re-scaling yields a "new" velocity field based on the following formulation:

$$u^{new} = u_*^{new} \left( \frac{u^{org}}{u_*^{org}} + \frac{1}{\kappa} \ln \frac{z_0^{org}}{z_0^{new}} \right) \quad (7)$$

where $\kappa = 0.41$, and the original flow field is denoted by superscripts "org" (Troldborg et al., 2022).

**Wind Farm**

The precursor flows are applied on the inlet boundary condition for the wind farm simulations, which has been performed on a new grid. The new wind farm grid is $L_X \times L_y \times L_z = 7687.68m \times 800.8m \times 800.8m$ corresponding to $192R \times 20R \times 20R$ in the streamwise, lateral, and vertical directions. The grid is equidistant from the inlet and in the vicinity of the turbines with a resolution of 20 cells per blade radius, and stretched towards the lateral, top and outlet boundaries. The resolution is quite high for actuator disc simulations, and the resolution is expected to give an error of less than $1\%$ in $C_T$ (Hodgson et al., 2021). The grid has $3392 \times 192 \times 128 \approx 83 \cdot 10^6$ grid cells. Cyclic boundary conditions are imposed on the lateral boundaries to mimic an infinitely wide wind farm.

The 14 turbines are spaced $12R$ apart in the streamwise direction and $20R$ in the lateral direction, due to the cyclic boundary conditions. The modelled turbine is the NM80 turbine, which has a radius of $R = 40.04m$, hub height of $z_0 = 80m$ and re-





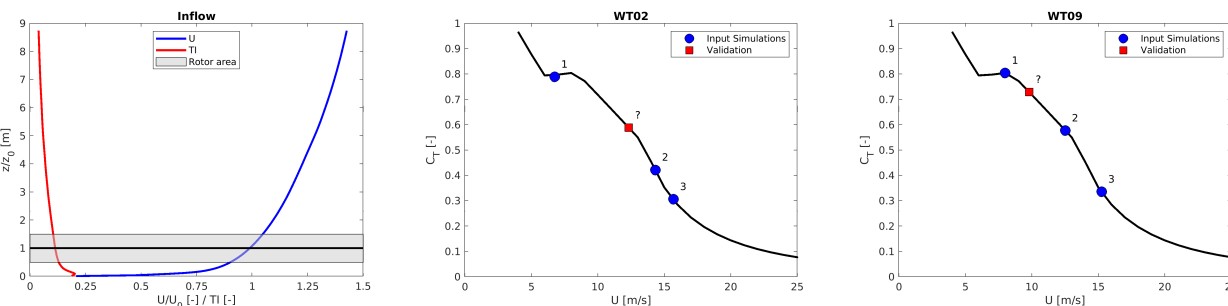

**Figure 2.** a) Normalized and averaged streamwise velocity and turbulence intensity profiles, b) Steady state $C_T$-curve and $C_T$-values for the wake generating 1st turbine and c) for the wake generating 8th turbine in the input simulations and validation case in blue and red, respectively.

scaled to rated wind speed of $U_{rated} = 14 m/s$ with corresponding rated power of $P_{rated} = 2,750 MW$ (Aagaard Madsen
et al., 2010).

**Flow cases**

The flow re-scaling is utilized in order to create a database, which covers a significantly reduced parameter space, where the thrust coefficient $C_T$ is the only governing parameter. The inflow is re-scaled to have the same turbulence intensity and shear exponent for all flows, while only the inflow velocity at hub height varies, and gives rise to differences in $C_T$ for $WT02$ and
$WT09$. The roughness ($z_0 = 0.051m$) is calibrated to give an average shear exponent over the rotor area of $\alpha = 0.14$, which implies a turbulence intensity of $TI \approx 11\%$ at hub height for all simulations, as all turbulence is mechanically generated for a neutral boundary layer. The friction velocity is calibrated to give inflow hub height velocities of $U_0 = 8 m/s, 15 m/s$, and $20 m/s$ at the position of the first turbine. Additionally, a validation simulation has been run with inflow hub height velocity of $12 m/s$.
The normalized inflow velocity and turbulence intensity profiles are shown in Figure 2a), which also shows the steady state $C_T$-curve of the NM80 in Figure 2b)-c). The average $C_T$-values of the 1st and 8th turbine for the three input simulations are marked with blue circles, while the validation simulation is marked by a red square.

The flow is initially developed throughout the wind farm for $1,800 s$ to remove any transients from the simulations. All cases have been simulated for $13,107 s = 3.64$ hours. The long simulation time is necessary in order to capture low frequencies and
to generate a significant amount of data for comparing the stochastic generation. A visual impression of the highly complex and turbulent wind farm flow is given in Figure 3, which shows streamwise velocity and iso-surface of the vorticity. The three velocity components $u, v, w$ are extracted in planes $1R$ upstream each turbine (as indicated) every 10Hz corresponding to $2^{17} = 131,072$ snapshots. The extracted planes cover $\pm 1.1R$ in the lateral and vertical. Here, the inflow to turbines 2 and 9 will be used to build PS-ROM, *i.e.* the wake generated by the 1st and the 8th turbine. The two cases showcase the generic capabilities
of the model to capture both single wake dynamics and the inflow to the 9th turbine is arbitrarily taken as representative of deep





wake conditions, where the statistical moments have generally converged (Andersen et al., 2020). Instantaneous snapshots of the streamwise velocity fluctuations are shown in Figure 4, where the dynamics of the turbulent wake are clearly evident.

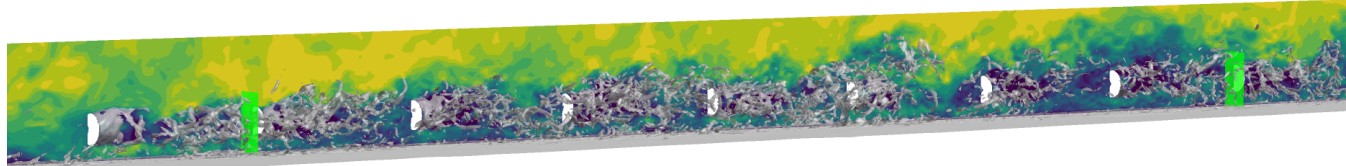

**Figure 3.** Vertical plane of streamwise velocity as well as iso-surface of the vorticity through a wind farm. Turbines are marked in white circles, and planes for extracting inflow velocity components for WT02 and WT09 are shown in green.

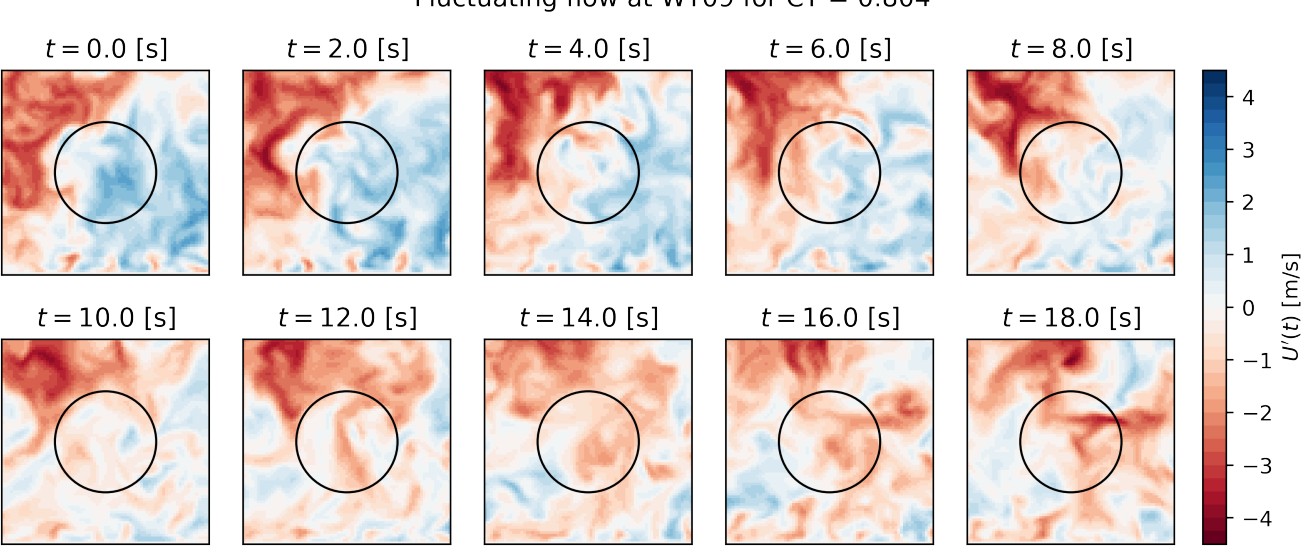

**Figure 4.** Instantaneous snapshots of the streamwise velocity fluctuations in a wake with $C_T = 0.806$.

Table 1 summarizes the inflow parameters for the wind farm simulations and the operational conditions of the wake generating 1st and 8th turbine. The validation case is chosen to show the predicting capabilities of the model, as the $C_T$-values of the two validation cases are located almost midway between two of the input simulations, *i.e.* the largest distance from simulated flows in the database. This is the case for both WT02 and WT09, although the distance from the validation cases to the input cases differ.



**Table 1.** Summary of inflow parameters for wind farm simulations and operational conditions of the wake generating 1st and 8th wind turbines.

| Simulation | $U_0[m/s]$ | $C_{T,1st}$ | $C_{T,8th}$ |
|---|---|---|---|
| 1 | 8 | 0.789 | 0.805 |
| 2 | 15 | 0.422 | 0.578 |
| 3 | 20 | 0.306 | 0.338 |
| Validation | 12 | 0.589 | 0.730 |

## 4.2  Global POD Modes and Modal time series

The global POD modes are computed using the database of inflows, and the first 15 modes for the 9th turbine are shown in Figure 5. Clearly, large structures are visible, which gradually diminish in size with increasing mode number, *e.g.* the monopole structure in mode 1 (and 9,11). Overall, these global and spatial POD modes are very similar to previously reported POD modes based on individual scenarios of either single wakes (Sørensen et al., 2015; Bastine et al., 2018) or multiple wakes (Andersen et al., 2013). The global POD modes for the inflow to WT02 is very similar, but not shown for brevity. This clearly indicates how the dominant coherent structures are similar across cases and yields the potential for deriving a ROM based on such generic building blocks.

However, the importance of each mode will differ from case to case. The original non-normalized flows are therefore re-projected into the global POD modes to obtain modal time series of each mode for each $C_T$-value. The time series are shown for the first five modes of both WT02 and WT09 in Figure 6. Low frequency fluctuations are predominantly visible for the first mode for both WT02 and WT09, and to some degree in the second modes, but effectively disappears for higher modes. Combined with the monopole structure of the global POD modes, this shows how the first mode is mainly a low frequent correction to the mean velocity profile. Overall, the amplitude of the modal time series decrease for increasing mode number for all three cases. This can also been seen in Figure 7, where the variance of the modal time series are plotted. The variance is a proxy of the turbulent kinetic energy in the modes, and the decreasing trends is very similar to the standard decrease in energy content of the POD eigenvalues. Some differences arise, particular between the single wake inflow to WT02 and the deep array wake at WT09. The contribution of variance is higher across all modes for the high $C_T$ for WT02, which could indicate that more modes are required to fully capture the flow physics. The variance across modes are very similar for the smaller $C_T$-values. This trend is also seen for WT09, where the variance is comparable, although slightly lower for the lowest $C_T$.

Turbulent wake flow is highly non-linear, but POD is a linear combination of spatial modes and its corresponding modal time-series. Therefore, non-linearity of the flow might emerge through the modal time series and their interaction, for instance correlation. Figure 8 show the correlation of the temporal coefficients of the first 15 POD modes for WT02 and WT09, where the diagonal is obviously unity. Generally, the off-diagonal correlations are very small. However, several interesting aspects become apparent. There are clear modal interaction in the modal time series. The correlations tend to transition as $C_T$ changes,





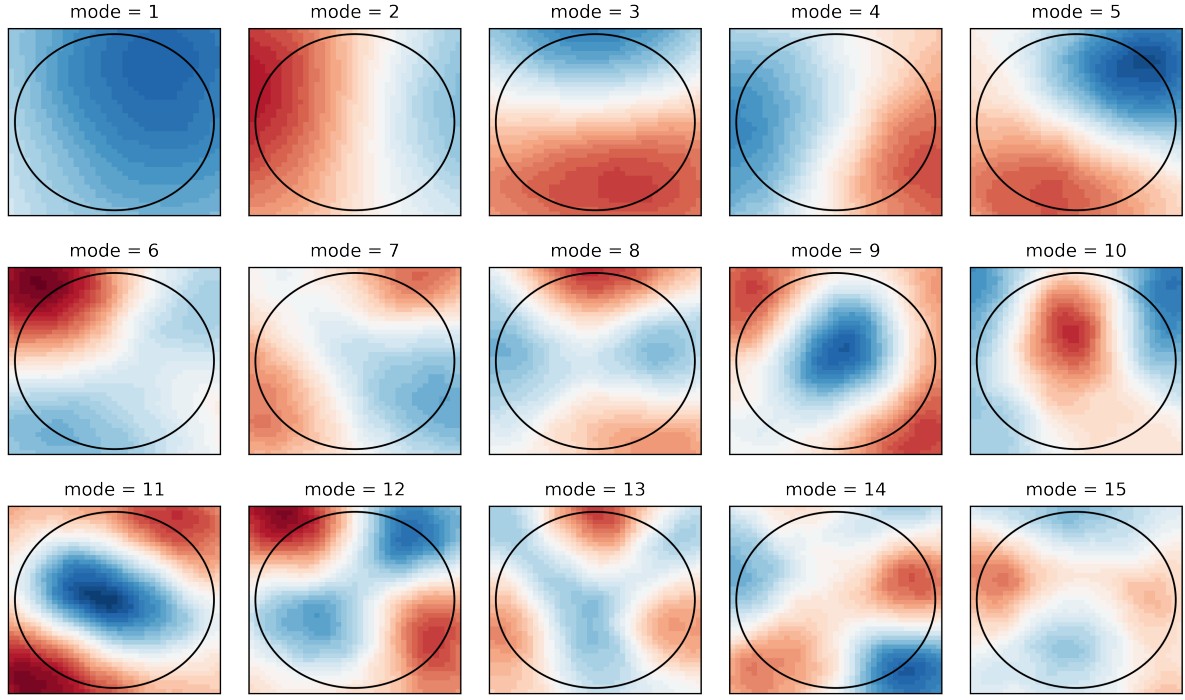

**Figure 5.** First 15 global POD modes, $u$-component, note that each mode also has $v$ and $w$ fluctuating velocity components.

*e.g.* modes 1-2 are negatively correlated for the lowest $C_T$ of WT09, uncorrelated for the intermediate, and positively correlated for high $C_T$. Similar trends are observed for WT02. However, for WT02 the correlations are generally higher for modes 5 and above, while the modal time series are mainly correlated for the first five modes for WT09. Interestingly, the correlations are basically non-existing for the intermediate $C_T$-value for both WT02 and WT09.

### 4.3 Cross Spectral Density and Stochastic Generation

The temporal interaction of the different modes can also be examined through the Cross Spectral Density (CSD), which represents the covariance between multiple time series across the frequency spectrum. The CSDs are complex, but the norm of the CSDs are shown in Figure 9 for the first two modes. The CSDs are estimated using Welch's method (Welch, 1967) with a moving Hann window of $2^{16} = 65,536$ time steps.

The different $C_T$-values affects both high and low frequency content across modes, and the kinetic energy is less for high $C_T$ over the entire spectrum as expected due to the lower freestream velocity. The energy content also decrease as the mode number increase. Some low frequency interaction is generally present for the shown modes, although it is altered significantly for the high $C_T$ in deep array. The norm of the CSDs are otherwise very similar across mode numbers.





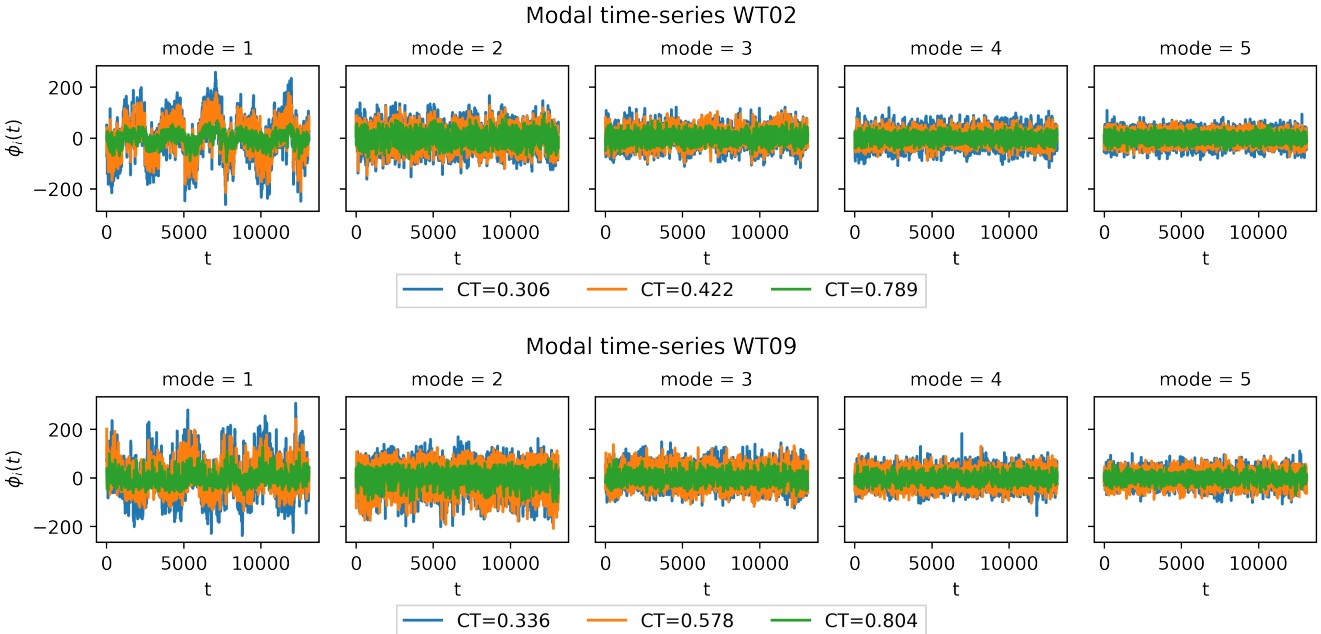

**Figure 6.** Modal time series of the first five global POD modes for WT02 and WT09.

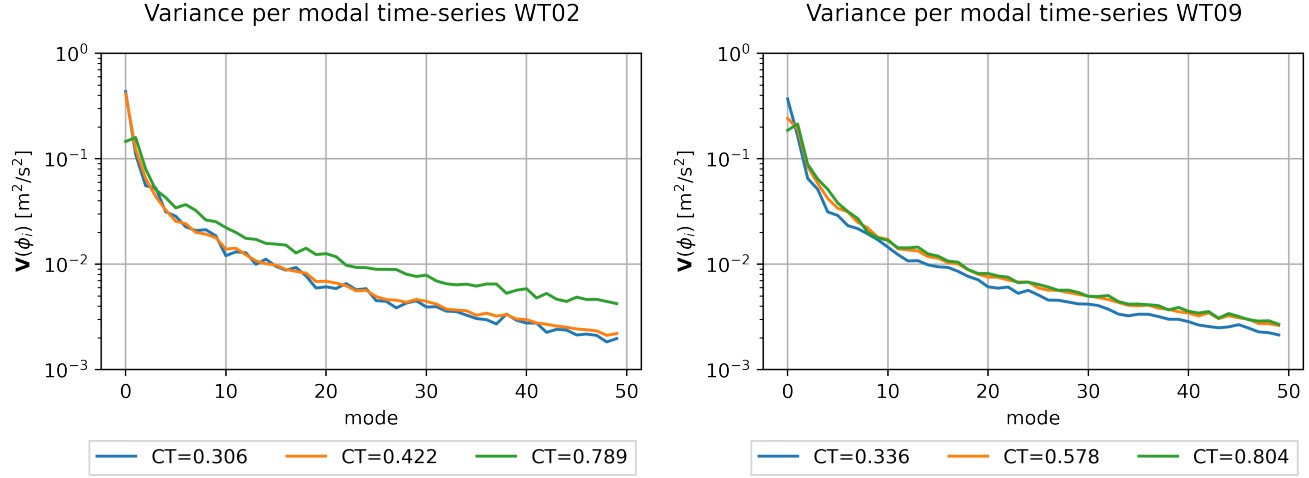

**Figure 7.** Variance in each modal time series for WT02 and WT09.

Stochastic multi-variate time series are generated from the complex-CSD. Figure 10 compares the original CSD norm with the CSD norm of 20 randomly generated multi-variate time-series. The resulting CSDs are clearly very similar to the original,

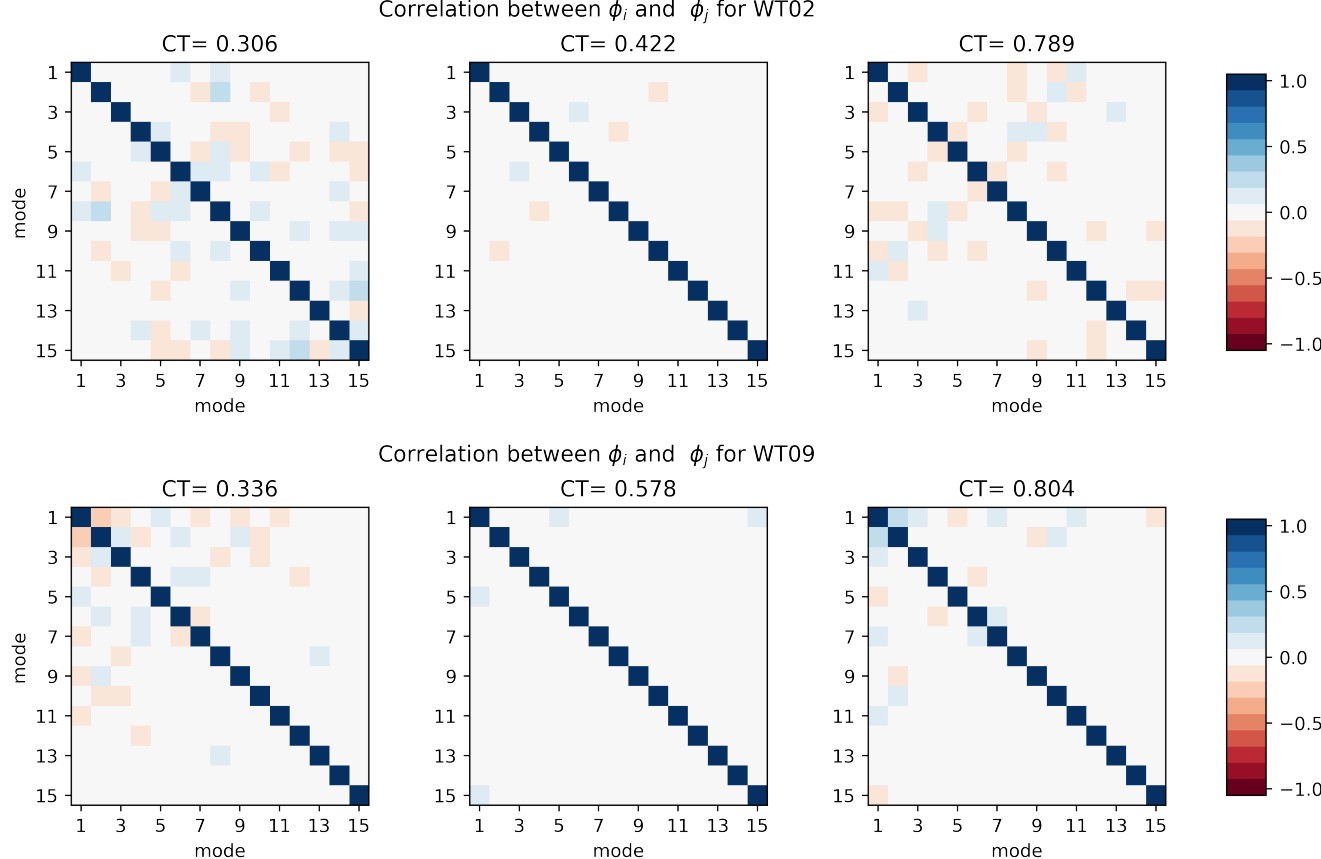

**Figure 8.** Correlation of the modal time series of first 15 global POD modes for the three $C_T$-values for WT02 and WT09.

for all modes and across the entire spectrum. Some spread from the random realizations are seen, which is expected and
255 desirable.

The histograms of the original modal time series and of the 20 random realizations of the time series are compared in Figure
11. The histograms are overall in excellent agreement, again showing that the random realizations capture the statistics of the
original signal. The histogram are generally Gaussian for WT02, while the original histograms are slightly skewed for the first
couple of modes for WT09 compared to the normal distributed random time series.

260 **4.4 Predicting out-of-sample flow cases**

The stochastic capability of the reduced order model needs to be supplemented by predictive capability for unseen cases,
*i.e.* predicting wake dynamics for $C_T$-values different from the three input simulations in Table 1. Here, a validation case is
included with $C_T$-values between simulation 1 and 2.



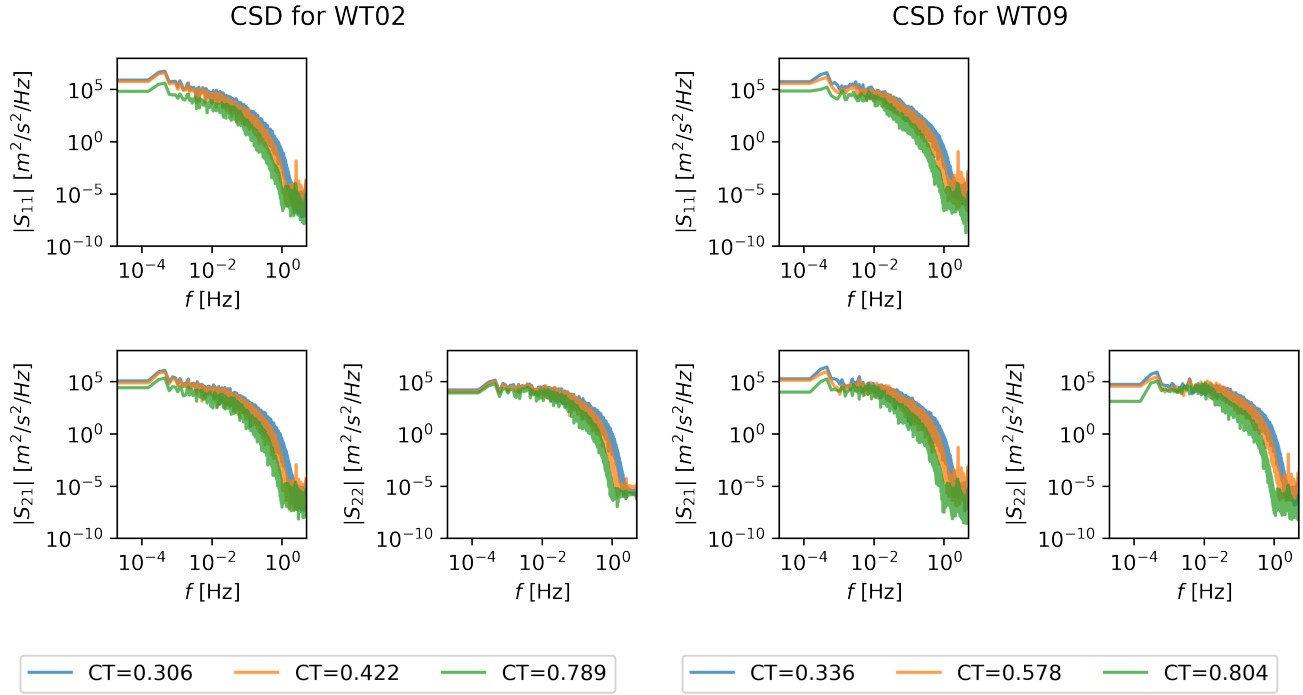

**Figure 9.** Norm of Cross Spectral Density of modal time-series of first two global POD modes for three $C_T$-values and for WT02 and WT09.

Stochastic realizations of the multi-variate modal time-series are predicted by interpolating the CSD at an out-of-sample
$C_T$-value. Here the CSD-surrogate, $\hat{\mathbf{S}}(f,\theta)$, consists of a direct interpolator of CSD as a function of $C_T$, see Figure 9, which
clearly shows the dependency of spectral variance on $C_T$. Figure 12 shows 20 random realizations compared to the validation
data. The agreement is excellent across the entire spectrum and for all mode numbers. Again, the random realizations show a
spread around the validation CSDs.

The dynamic wake inflow can now be reconstructed for each realization of the modal time-series to compare with the
validation simulation. Figure 13 show a random stochastic realization using 50 modes, where the dynamics are clearly seen for
different instantaneous snapshots.

### 4.5 Model Validation

The model is validated by comparing results of aero-elastic (Flex5) computations using both the original LES inflows and
100 random realizations generated by PS-ROM. The generated time series are $13,107 s$ long for both the original LES and the
PS-ROM realizations. The realizations are split into periods of $650 s$, which has an overlap of $300 s$. Therefore, the LES dataset
contains 42 statistical samples, while the 100 random seeds of PS-ROM yields a total of $4,200$ samples. Statistics are based
on $10 min$, including mean power and 1Hz damage equivalent loads (DEL) for the main load channels of blade root flapwise

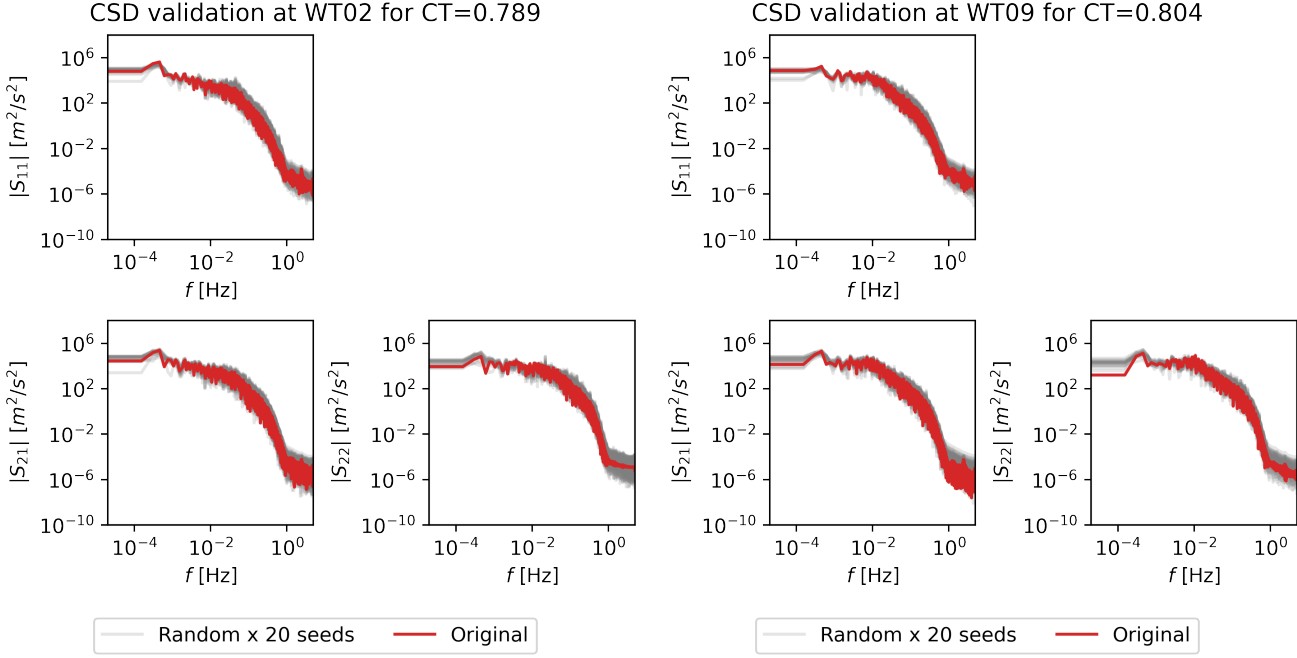

**Figure 10.** Comparison of CSD of original and stochastic realizations evaluated at high $C_T$ (a member of the "training" dataset) for WT02 and WT09.

bending moments (BRF), tower bottom fore-aft bending moment (TBF), and tower bottom side-side bending moment (TBS). The turbine will effectively act as a spatial filter, so higher modes with small scales can eventually be neglected.

280    Figure 14 shows violin plots of mean power and 1Hz DEL of BRF, TBF, and TBS for PS-ROM realizations using 5, 10, 20, and 50 modes compared against the original LES inflow (red) for both WT02 and WT09. Generally, the distributions from PS-ROM are Gaussian and it is clearly seen how the mean and width of the distributions increase as more modes are added to the flow generation, particular for the DELs. It is expected that the stochastic generation of PS-ROM will result in wider distributions for most quantities as there are 100 times as many realizations, provided that sufficient number of modes are used

285    for the flow generation. Importantly, the distributions are clearly bounded.

The median and quartiles (25% and 75%) of power are captured very well by PS-ROM for both turbines, although the LES validation dataset is non-Gaussian and with two distinct peaks. The PS-ROM distributions and therefore the extreme percentiles (5% and 95%) are wider in power for WT02 than for validation LES dataset, but very comparable for WT09. Similarly, the distributions of BRF are also well-captured, although it requires more than 10 modes for WT02 and more than 20 modes for

290    WT09. The distributions of tower DELs are initially more sensitive to adding more modes, particular the tower bottom bending moments. Adding more modes significantly increase the median and width of TBF and TBS, so they match the validation distributions very well for more than 20 modes. TBF for WT09 show the worst comparison, where the PS-ROM realizations



WIND
ENERGY
SCIENCE
DISCUSSIONS

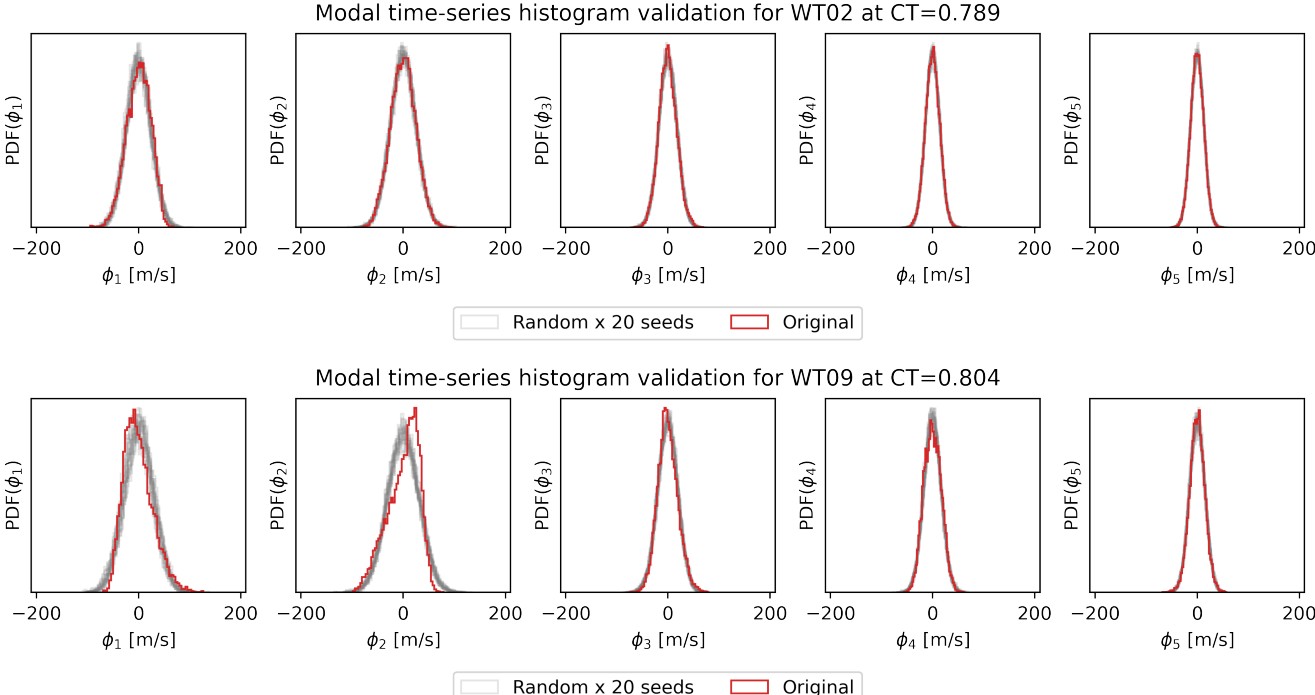

**Figure 11.** Comparison of modal time-series histograms for original and random realization evaluated high $CT$ (a member of the "training" dataset) for WT02 and WT09.

underpredict compared to the LES validation distribution. However, it is noteworthy how the distributions from LES are very wide for WT09, arguably due to the increase turbulent wake dynamics in the deep array, which might require a lot more modes.

295     Overall, the statistical comparison is very good, and it appears that 20 modes or more is generally sufficient to capture the distributions of power and the dominant loads. Obviously, the turbine acts as a spatial filter, which makes the flow generation more efficient as fewer modes are required (Saranyasoontorn and Manuel, 2005; Andersen, 2013). It is substantially fewer modes than previously reported by Andersen (2013), although the stochasticity helps to yield the correct statistical distributions.




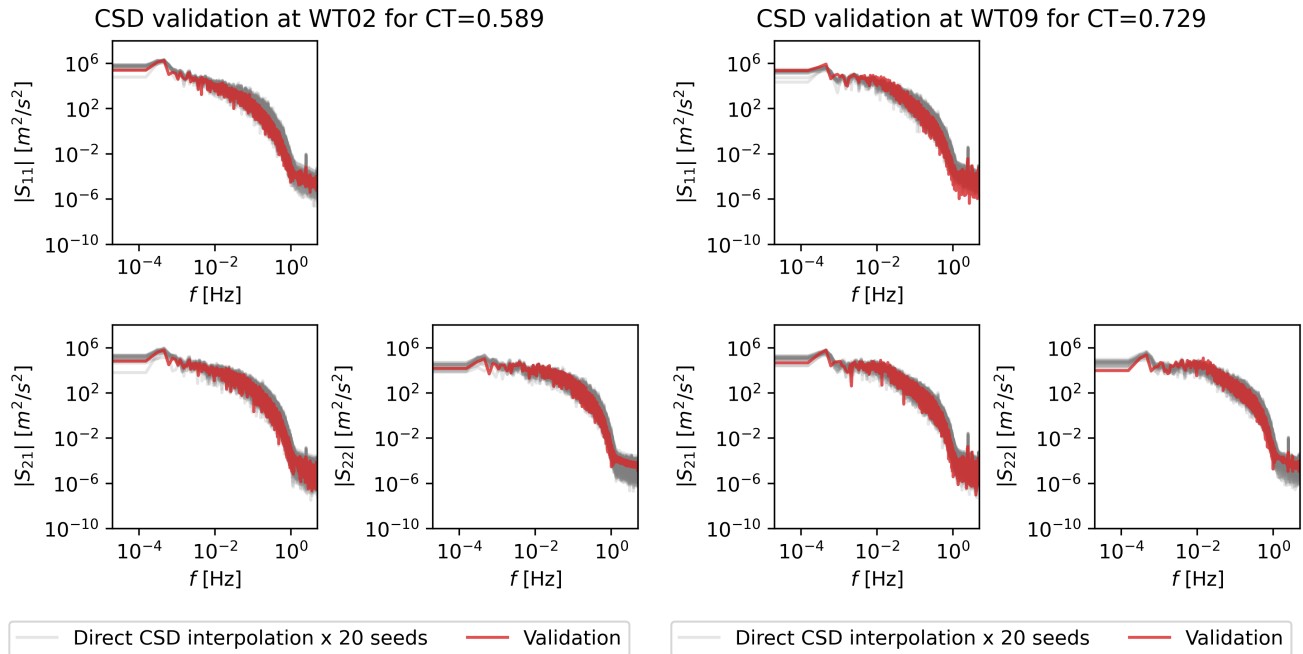

**Figure 12.** CSDs of 20 realization of the PS-ROM using direct interpolation compared to the validation CSD for WT02 and WT09.

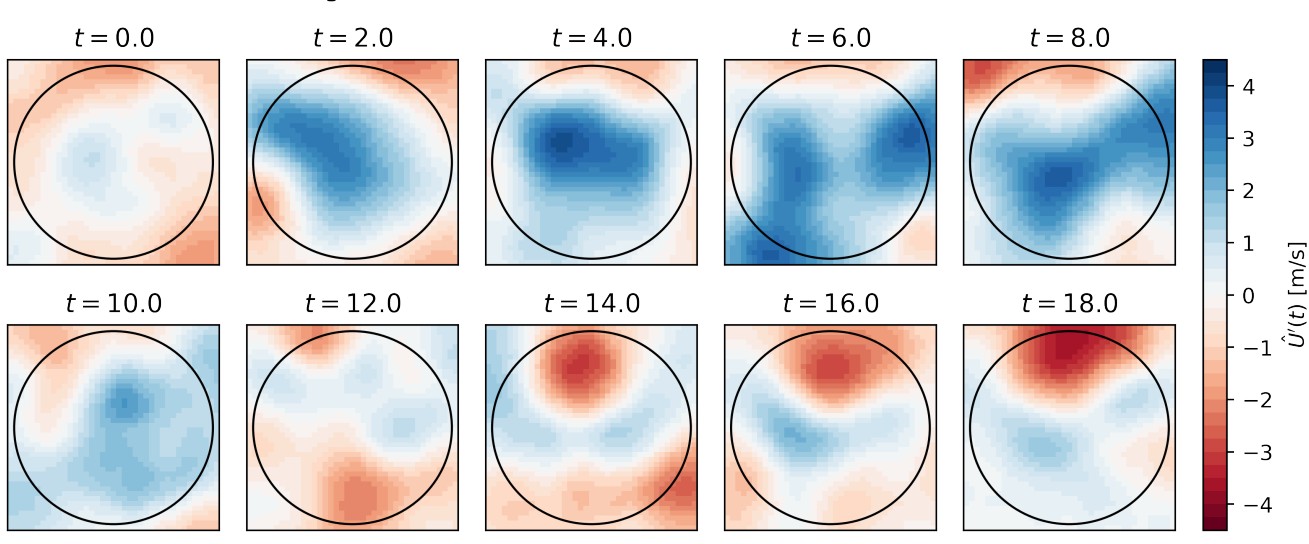

**Figure 13.** Instantaneous snapshots of the stochastic realizations of the streamwise velocity in a wake with $C_T = 0.729$ predicted with PS-ROM using 50 modes.

**Figure 14.** Distributions of $10min$ mean power and 1Hz damage equivalent load (DEL) of blade root flapwise (BRF), tower fore-aft (TBF), and tower side-side (TBS) bending moments for LES and for PS-ROM with 5, 10, 20, and 50 modes. Prediction at validation case, not used in the training of PS-ROM. Horizontal lines corresponds to $5, 25, 50, 75$, and 95-percentiles. The statistics from PS-ROM is based on 100 random realizations of time series of $13, 107$ [s] long corresponding to $2^{17}$ flow snapshots. WT02 is shown on the left and WT09 on the right.





# 5  Discussion

## 5.1  Assumptions, Limitations and Advantages

PS-ROM is initially intended for dynamic wake simulations in order to quickly assess power and loads on wind turbines operating in wake, which requires accurate and fast flow generation. The computational time of generating new time series of $2^{17}$ snapshots is approximately $1 - 2mins$ using a python implementation and depending on the number of global POD modes (and computing architecture (DTU Computing Center, 2021)), *i.e.* more than a 100 times faster than real time. For comparison, the execution time for generating a Mann turbulence box of comparable dimensions ($32 \times 32 \times 2^{17}$) is $6min$ for a Fortran implementation. DWM requires generation of two Mann boxes, filtering as well as combining the different submodels, so timing is significantly longer. Each LES simulations required a total of approximately $39,000$ cpu hours. Hence, PS-ROM is orders of magnitudes faster than the alternatives, while also providing statistical distributions including the extreme estimates with high fidelity LES accuracy.

The high accuracy is a significant advantage of PS-ROM compared to engineering models, such as DWM, with simplified physics and which are typically highly parameterized, and therefore require calibration, *e.g.* Reinwardt et al. (2021). A particular critical assumption of engineering wake models relates to the superposition of individual single wakes to obtain flow inside wind farms. Numerous methods exists (Porté-Agel et al., 2020), but the models are particular sensitive around rated wind speed, where models typically switch binary between different methods, although the changes in $C_T$ are continuous. This naturally affect the loads predictions of such models (Larsen et al., 2017). PS-ROM inherently captures multiple wakes and the change from one control-regime to another, *e.g.* from below rated to above rated wind speed.

PS-ROM also has certain limitations related to the assumptions of the model components. First, PS-ROM is dependent on Welch's method and the selected window size. The present results clearly shows how the method can capture low frequencies, but it requires sufficient data to resolve low frequencies and to satisfy the assumed ergodicity required for the multivariate Gaussian processes. Welch's method also averages the spectra, which could in principle limit the variability between realizations. Whether the stochastic generation will give rise to the same variability remains unknown, although the distributions indicate that the width of power and load distributions are captured very well. However, the clear non-Gaussianity of the original LES dataset is very challenging to capture and will require additional steps to generate non-Gaussian modal time-series, see Figure 11. This could have an impact in the distributions of power and loads, which are currently Gaussian as predicted by PS-ROM. Second, the presented implementation of PS-ROM only predicts the turbulent fluctuations, and therefore relies on knowing the mean flow across the parameter space. The mean wake flow can in principle be determined from an engineering wake model (*e.g.* Porté-Agel et al. (2020)), CFD-RANS (*e.g.* van der Laan et al. (2019)) or developing a surrogate of the mean flow based on the LES (*e.g.* Zhang et al. (2022)). Third, the flow fields are extracted $1R$ upstream the turbines, and therefore includes the effect of induction directly. Hence, PS-ROM does not incorporate changing control of the wake affected turbine, *e.g.* individual pitch control or yaw misalignment. However, the induction at $1R$ upstream is essentially independent of turbine specific details for the same $C_T$ (Troldborg and Meyer Forsting, 2017), so it is assumed to only have minor influence. The low frequency content of the generated turbulent fluctuations can in principle be corrected for induction (Mann et al., 2018) as





well as corrections to the mean velocity due to induction (Troldborg and Meyer Forsting, 2017). As the model is assumed to essentially be independent of the specific wind turbine model, it is also expected that the model can be scaled geometrically

for different turbine sizes operating at same $C_T$ (van der Laan et al., 2020).

    Scaling the model significantly expands the generality of PS-ROM, and the range of applications. As shown, PS-ROM is faster than DWM, and yields loads and power distributions with LES accuracy. Therefore, PS-ROM can be applied similar to DWM to generate load surrogates of wind turbines operating in wind farms (*e.g.* Dimitrov (2019)), and hence used for wind farm plant design. PS-ROM can also be used to improve control algorithms of individual turbine controllers to be specifically

designed to operate in wake dynamics as opposed to the present practice of tuning wind turbine controllers for free-stream conditions.

    PS-ROM is currently limited to cover only the one parameter space of $C_T$. However, the stochastic nature, LES accuracy, and predictive capabilities across the parameter space effectively implies that PS-ROM can be viewed as a generalization of LES for statistically stationary flows. This has immense impact for the application of high fidelity LES in general, and not only

for wind farm flows applications. Hence, PS-ROM can boost and enhance the value of a few LES beyond the case-specific cases, and thereby reduce the effective computational resources required for achieving high fidelity results of highly turbulent flows.

## 5.2    Flow Physics

PS-ROM enables a consistent approach to explore the underlying physics governing highly turbulent flows. POD has tradition-

ally been used to gain insights into the dominant coherent turbulent structures, and assigned as drivers of main flow features. However, such analysis have limitations in the sense that these structures can not be measured or seen independently, but only discovered though POD. Eventually, the higher mode structures merely represent an optimal mathematical basis of the turbulent fluctuation in space without specific physical interpretation besides small scale turbulence. PS-ROM provides global modes, which decrease in size with increasing mode number. The global POD modes clearly resemble spatial POD modes of

individual cases, and although they might be sub-optimal in terms of capturing the energy compared to individual cases they clearly show the power of using global POD modes across the parameter space to elucidate on the flow physics. Changes in the parameter space are captured in the modal time series, and significant transitions will effectively change the importance of the individual global POD modes. PS-ROM proves that it is possible to interpolate over relatively large distances in the parameter space to get the weight of individual global modes through the CSDs of the modal time series. This is opposed to previous

results indicating that information is only locally available, and can not be extrapolated globally (Christensen. et al., 1999), or by interpolating between modes derived from individual cases to capture transitions (Stankiewicz et al., 2017). Interpolation in the CSDs is conjectured to be more robust than interpolating in the transition between locally derived modes.

    Figure 8 shows how the modal interaction change in physical space, *i.e.* whether PS-ROM is build for WT02 or WT09. Clearly, the modal interaction is spread on a larger number of modes for the wake generated by the first turbine, where the

atmospheric flow is highly influential on the wake dynamics. Further in the farm at WT09, the modal interaction predominantly occurs between the lowest modes. The correlations are a direct manifestation of non-linearities in the flow, as it shows that





the modes are not independent. Therefore, the flow dynamics can not be separated and linearized, but they need to be solved collectively. Separation of scales and the associated linearization prevent methods from fully capturing the flow physics, *e.g.* DWM and various methods for Galerkin projection. Galerkin projection methods typically solve the Navier–Stokes equations

independently for each POD mode, which does not guarantee stable nor efficient results despite significant efforts to include the non-linearities (Xiao et al., 2019). Conversely, PS-ROM is a more empirical approach, but ensures that the inter-scale non-linear interactions are maintained in the generated turbulent flows. The influence of $C_T$ on the correlations between modal time series also offers interesting insights on transitions from low to high $C_T$ values. Surprisingly, the non-linearities seem to vanish for the intermediate $C_T$-values. This indicates that the flow is more linear as the modes are independent for flows, where

neither the atmospheric nor the wake dominates.

Another significant transition in the wind farm flow is seen in Figure 9. The low frequency content is visible for all $C_T$-values of WT02. However, for WT09 the low frequency content of mode 1 is significantly reduced for high $C_T = 0.804$. This corroborates the finding of Andersen et al. (2017), where the most energetic lengths scales are eventually seen to be limited by the turbine spacing, *i.e.* for high forcing the turbines can break up the largest atmospheric flow structures. The low frequencies

remain in the second mode, but significantly reduced for WT09 compared to WT02. Contrarily, it has implications for DWM, which assumes that the low frequencies of the atmosphere govern the low frequency meandering throughout the wind farm.

### 5.3 Future Developments and Open Questions

Future developments will extend PS-ROM to cover a multi-dimensional space in order to increase the application range even further. However, the parametric space is large, so the sparsity of LES dataset will present its own challenges to overcome

the "curse of dimensionality". It is an open question how to perform multivariate interpolation, while enforcing the necessary Hermitian properties of the CSDs. Despite sparsity of high fidelity datasets, the simulations might require insurmountable computational resources to derive the global POD modes. Alternatives to POD exist for deriving global spatial basis. Generic and theoretically derived modes, *e.g.* Fourier-Bessel functions, could also be used. Formally, these modes require the variables of the eigenvalue problem to be stationary or homogeneous (George, 1988), but such modes could in principle be used irre-

spectively. It would simplify the determination of the global basis, but would also enforce significant constraints in terms of assumed symmetries. Hence, it could inherently enforce an assumption that vertical shear and veer only have linear influence on the flow and could be removed by initially subtracting the mean. From Figure 5 it is clearly not the case as the first modes is asymmetric and appears to be a low frequent correction to the mean velocity. Alternative, could be to employ various machine learning dimensionality reduction methods, for instance auto-encoders, which presents a non-linear generalization of

POD (Hinton and Salakhutdinov, 2006). Autoencoders can build ROMs for complex nonlinear problems with higher accuracy compared to POD. However, autoencoders are computationally demanding and less general in the sense that the optimal autoencoder architecture could change every time a new parameter-scenario is added. PS-ROM does not suffer the same limitations due to its simplicity, both conceptually and in terms of model architecture, which is essentially a one-layer model. Nevertheless, the use of autoencoders could potentially increase the accuracy in PS-ROM.





Therefore, the fundamental model framework of PS-ROM is conjectured to be applicable for a wide range of highly turbulent flows, beyond wind farm flows. As PS-ROM is data-driven it only requires high fidelity data resolved in time and space, and the model framework can therefore also be applied to large measurement databases.

## 6   Conclusions

A predictive and stochastic reduced order model for turbulent flow fluctuations in wind turbine wakes is presented. The model
is constructed using a high fidelity LES database of the turbulent inflow to turbines operating in a large wind farm. The thrust coefficients of the wake generating turbines is the governing parameter of the flows. A set of global building blocks are derived by applying proper orthogonal decomposition to the combined database covering the parameter space. The modal time series are obtained by reprojection the original flow into the global modes. The modal time series and corresponding cross spectral density show clear dependency on $C_T$, which enables a direct interpolation in the CSDs. PS-ROM is therefore able to predict
unseen cases, and generate random stochastic realizations with the correct spectral statistics. The model is validated against an independent LES in terms of power and damage equivalent loads. PS-ROM yields very good agreement in the distributions, including the extreme estimates of $5\%$- and $95\%$-percentiles.

A validated, fast and truly dynamic wake model have a number of applications:

–   Full load distribution estimation. The fast creation of new realizations will enable to cover the full load distribution for a
given operating condition, *i.e.* achieve actual estimates of extreme loads, *e.g.* P95 estimates.

–   New control algorithms dedicated for turbines operating in wake. Advanced controllers can be designed to include different operation depending on e.g. turbulence intensity Dong et al. (2021), but extensions could be made to incorporate the specific time and length scales of wake dynamics as captured by the present model.

–   Wind farm optimization including loads by using PS-ROM to construct load surrogates.

The proposed model also provides significant insights to the physics of highly dynamics wind farm flow. It reveals how the non-linear interaction of the global modes change significantly for different values of $C_T$, where non-linearities are mainly present for low and high values of $C_T$. The high values of $C_T$ is also seen to significantly alter and reduce the atmospheric low frequencies in the deep array. The predictive and stochastic nature of the reduced order model framework can been seen as the generalization of LES beyond the case-specific cases. Additionally, the consistent physical modelling and analyses are
expected to be generally applicable for statistically stationary and highly turbulent flows, not only wind farm flows.

*Video supplement.*   An animation of a stochastic realization using 5, 10, and 50 modes is available.



*Author contributions.* Idea and conceptualization (JPM, SA), LES and Flex5 simulations (SA), model implementation (JPM), analysis and paper writing (SA,JPM).

*Competing interests.* No competing interests.

*Acknowledgements.* The work has partly been funded by DTU Wind Energy through the Wind Farm Flow CCA 2022, and by Nordic Energy Research through the funding of Interconnecting the Baltic Sea countries via offshore energy hubs (BaltHub, project number 106840). Computational resources has been provided by the DTU cluster Sophia. DTU Computing Center (2021).



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
