# Peer review of "Predictive and Stochastic Reduced Order Modelling of Wind Turbine Wake Dynamics"

_Wind Energy Science, 2022_

## Author Comment (AC1)

**Response to Review:**
**Predictive and Stochastic Reduced Order Modelling of Wind Turbine Wake Dynamics**

Thank you for the detailed review and the positive comments. We have addressed all comments in the updated manuscript and below, where review comments are given in black and response in blue. It should be noted that some minor updates have occurred. We have replaced Welch with a logarithmic-window moving average filter to reduce the spread of the spectra without loosing low frequency information. This improves the spectra, particular at low frequencies because it can better resolve lower frequencies and removes noise at high, which improves interpolation. We have also decided to plot Figure 11 with log-scale on the y-axis to provide clearer comparison of the histograms.

**Response to Reviewer 1:**

The authors present a predictive and stochastic reduced-order model based on proper orthogonal decomposition for wind turbine wake flow application. The authors did an excellent job explaining the methodology, which I expect will be used by others in the field. The paper should definitely be published. I just have a few minor comments and requests for clarifications.

Thanks your review and positive comments, please see our response below.

1. The introduction section does not provide recent state of art in using POD and/or ROM for wind energy applications. Also, some of the motivations of the current work are already resolved by other studies. For example" Cluster-based probabilistic structure dynamical model of wind turbine wake", and "Clustering sparse sensor placement identification and deep learning-based forecasting for wind turbine wakes". There have been many more studies that should be mentioned here.
We have of course only provided a general overview of what has previously been done in the field, and we are aware of these particular articles. The first does indeed attempt to solve the same challenge as our work, and we have now included it as a reference. We recognize that the Markov Chain and clustering approach yields stochastic realizations of the modal time series, but with a very coarse discretization due to the number of clusters (7). The method we are proposing is equivalent to using infinite clusters for the cluster-based probabilistic ROM. The transition matrix with infinite cluster should contain the same information as the cross-spectral density matrix. Therefore, it is not immediately evident from Ali et al. whether their proposed method is actually able to produce time series with the accurate cross-spectra properties, as such tests are not performed. As discussed and shown in our article, these properties are essential. Additionally, the suggested references does not include the predictive capability beyond the input simulation used to derive the reduced order model. We have also included a reference to Qatramez and Foti, 2022, which attempts to do similar things and have a good discussion on why this problem is difficult.

2. Can you provide details on the time resolution considered in evaluating the POD? How did the authors avoid the correlated features in POD calculation?

The other reviewer also asked about this, and we have now clarified the time resolution of the POD, where every 100th snapshot was used, i.e. every 10sec. The correlation between snapshots is essentially unimportant, because we effectively oversample the flow. We have paid closed attention at having long enough simulations to capture the low frequency components of the flow and enough snapshots in the global POD calculation to have all scales covered by the reduced order model, this can be confirmed by the fact that the last eigenvalues are basically zero. Furthermore, we wish to remind the reader that the important thing here is to obtain a representative and good basis, but the obtained basis will not be "optimal" in the classic POD sense. The proper temporal correlation is obtained afterwards as the flow is projected into this basis.

3. Figure 7: It is helpful to add the energy profile of the POD eigenvalues to the figure to visualize the trend of both.
Thanks for this comment. As addressed to the other reviewer, the eigenvalues are essentially nonsensical, and we believe such a figure would create more confusion than clarification of the method. We acknowledge that this is one of deviations from classical POD analysis.

4. Fig 8: Can you explain why cases CT=0.422 and CT=0.578 showed less correlated Eigenvectors than the other cases?
Figure 8 shows the correlation between the time series obtained from projecting the flow into the global basis. Our interpretation of the reduced correlation is as written in the Discussion that these intermediate flows are neither dominated by the atmospheric background flow nor the turbine generated wake.

5. Fig 13: Can you provide details on the energy content in these 50 modes?
The associated variance effectively yields the energy content of the modes, and hence the relevant information is shown in Figure 7. However, since the modal time series are correlated it is not possible to simply do a linear sum to obtain the energy content.

6. It is very practical of the authors to explain the limitations of the current work. Please explain your insight on how to develop the current approach to capture the non-Gaussian trend and if we have yaw/ tilt mechanisms.
Non-Gaussian time series can in principle be generated using an iso-probabilistic transfer function from the Gaussian to non-Gaussian distributions. However, as the validation shows and as discussed this might be of minor importance for power and load distributions.
The proposed method is completely generic, so it can also be applied for yawing or tilting turbines, where e.g. yaw angle would be the parameter in question for constructing a global basis. Yawing a turbine naturally leads to a change in CT, so it might require a two-parameter implementation. This is ongoing work in our group.

---

## Author Comment (AC2)

**Response to Review:**
**Predictive and Stochastic Reduced Order Modelling of Wind Turbine Wake Dynamics**

Thank you for the detailed review and the positive comments. We have addressed all comments in the updated manuscript and below, where review comments are given in black and response in blue. It should be noted that some minor updates have occurred. We have replaced Welch with a logarithmic-window moving average filter to reduce the spread of the spectra without loosing low frequency information. This improves the spectra, particular at low frequencies because it can better resolve lower frequencies and removes noise at high, which improves interpolation. We have also decided to plot Figure 11 with log-scale on the y-axis to provide clearer comparison of the histograms.

**Response to Reviewer 2:**

The article provides a model derived from LES to predict turbulent wind turbine wake dynamics, applied at two downstream locations within a long row. The model is built in a reduced parameter space denoting that the thrust coefficient indirectly encapsulates the effects of numerous other parameters of the flow, such as atmospheric turbulence intensity, spacing of the farm, etc. Proper orthogonal decomposition is employed to build the predictive and stochastic model.

Reasoning for why the model is relevant to the community and the general description of the methods are well described in the article. I believe the article is of importance and can benefit future turbulence modeling and recommend the article for publication with revisions based on the points outlined.

Thanks for the detailed review and the positive comments. We have addressed all comments in the updated manuscript and below. Following your comment we have decided to remove the sentence on indirectly encapsulating spacing etc for clarity.

Specific comments:

Figure 2a – I believe the y axis (z/z0) should be nondimensional.
Thanks for noticing. We have updated the figure.

It is unclear how many snapshots are used for the POD, is it all 131,072 at 10Hz?
We agree that we should have written this more clearly. We have now specified how every 100th snapshot is utilized for the global POD.

Why are 50 modes used for reconstruction? How much energy is associated with those 50 modes?
50 modes is to some degree arbitrary. We have used 5, 10, 20, and 50 to show the convergence and sensitivity of including more modes. Essentially, we do not gain much information by including more than 50 modes, when the aim is to capture power and load distributions. But there is no limit to how many modes can be included. In terms of the associated energy, see elaborate answer

below related to the variance and eigenvalues.

Why do the authors present the variance of the model time series in lieu of the eigenvalues? It is noted that the decreasing trends are similar but do the trends between the cases show similar results as well? For the near and far-downstream turbines?
The global eigenvalues are essentially nonsensical as we utilize a global POD, where we combine datasets. As we project the flow into the global POD modes we obtain the associated modal time series, and the resulting variance corresponds to the energy content. However, we can not sum the variance of the modal time series to obtain the "full" energy content as the time series are correlated. To summarize, Figure 7 highlights two aspects:
1. Overall, the decreasing trend show that the although the global basis is not fully optimal for the individual cases as a single POD would be, it is very close. This is part of the strength of the model and the reason, why it works across very different flow scenarios.
2. The (minor) "non-optimality" is occassionally seen as the variance of higher modes is larger than the variance of smaller modes, i.e. for certain flow cases the global modes effectively change places in terms of importance.
We have now expanded this explanation further in the text.

Is there any reasoning for the skew of the PDFs for the first few modes of the farther downstream turbine and in turn, why this is not captured in the model?
Good question, which is harder to fully answer. It is very interesting that this only happens for the lowest modes, which are related to the low frequencies and which we associate with corrections to the mean flow. It makes sense that a time series that corrects the mean flow could have skewed distributions. This is not captured by the model, because the spectral time series generation only yields Gaussian marginally distributed time series of the modes. However, that does not imply that the constructed flow time series are Gaussian. The correlation across modal time series introduces non-Gaussianity in the constructed flow.

Specific corrections:
Thanks, we have implemented all the following suggestions.

Line 20: captures – > capture

Line 22: particular – > particularly

Line 62: of model by – > of the model by

Line 229: arise, particular between – > arise, in particular between

Line 248: values affects – > values affect